# Analysis of Direct Optical Ablation and Sequent Thermal Ablation for the Ultrashort Pulsed Laser Photo-Thermal Micromachining

**Chang-Wei Xiong [1],[†], Ching-Yen Ho [2],[3],[*],[†] and Dong-Kai Qiao [2],[*]**

[1]  Department of Mechanical and Electrical Engineering, Dongguan Polytechnic, Dongguan 523808, China; xiongcw@dgpt.edu.cn
[2]  School of Mechanical and Electrical Engineering, Guangdong University of Petrochemical Technology, Maoming 525000, China
[3]  Department of Mechanical Engineering, Hwa Hsia University of Technology, Taipei 235, Taiwan
[*]  Correspondence: hcy2126@cc.hwh.edu.tw (C.-Y.H.); mmcqdk@gdupt.edu.cn (D.-K.Q.); Tel.: +886-961-107-053
[†]  These authors contributed equally to this work.

**Abstract:** An ultra-fast pulsed laser for materials processing can obtain submicrometer- to nanometer-sized parts or patterns (precision or accuracy) because the heat cannot diffuse in time for an ultra-fast pulsed duration, and this causes a threshold of ablation in multi-photoabsorption. The optical and thermal effects significantly affect the processing quality of an ultrashort pulsed laser for materials. This study utilizes a Laplace transform method to display the optical and thermal effects on the temperature field and the ablated depth of an ultrashort pulsed laser processing of materials. The results reveal that If an ultrafast pulsed laser-induced heat can keep the irradiated region above the evaporated temperature until the thermal diffusion occurs in the lattice of materials, thermal ablation occurs. The optical ablation can get a better processing quality due to less thermal diffusion. This study theoretically elucidates that the depth of optical ablation approximates the product of an optical absorption length and the logarithm of the ratio of laser fluence to laser fluence threshold. It has also been shown that the optical and thermal ablation, respectively, occur in low and high laser fluence because the optical ablation depends directly on the main source of the incident ultrashort pulsed laser. However, the thermal ablation is determined by the residual heat directly from the incident ultrashort pulsed laser after the optical ablation. The increase rate of the ablated depth per pulse with laser fluence is actually lower at high laser fluences than that at low laser fluences because the thermal ablation using the residual heat directly from the incident ultrashort pulsed laser is governed at high laser fluences. This study will provide the closed-form of a solution that elucidate the direct optical ablation and sequent thermal ablation for the ultra-fast pulsed laser photo-thermal processing.

**Keywords:** ultra-short pulsed laser; optical ablation; photo-thermal processing

## 1. Introduction

Lasers can be applied in many machinings of materials such as laser heat treatments [1], drilling [2], cutting [3–5], welding [6–8], hardening process [9], polished surfaces [10], nanometer-scaled processing [11], etc. The nanometer-scaled processing was generally achieved by an ultrashort pulsed laser. An ultrashort pulsed laser has the characteristics of the general laser, such as contactless processing, wear-free, clean surfaces, flexible arrangements of beam deflection and sample positioning, etc. It can conduct machining almost for all materials. An ultrashort pulse laser can give an ultrahigh

intensity and ultrahigh power source for laser material processing due to the ultrasmall focal spot and ultrashort pulse duration. Therefore, an ultrashort pulsed laser can be applied in laser micro- and nano-scaled processing of materials. The main advantage of an ultrashort pulsed laser is to get the smooth machining surface due to the ultrashort pulsed duration of insufficient thermal transport time to other regions of machining locations. Apart from longer pulses (>10 ps), an ultrashort pulsed laser processing of materials can reduce thermal damage to a minimum. As a result, the micro- and nano-scaled precise processing of materials are achievable during the ultrashort pulsed laser process [12].

Polymers [13–16], ceramics [17], glass [12,18–20], metals [21,22], diamonds [23], sapphire [17], etc. can be processed with an ultrashort pulsed laser. The nonmetals such as polymers, ceramics, diamond, sapphire have no free electrons, which make them exist in different physical ablation characteristics of metals with free electrons. For metals, the energy of an ultrashort pulsed laser is initially absorbed by the electrons during ultrashort laser pulse, and then the electrons translate heat to phonons in the lattice after several picoseconds. For nonmetals, the electrons absorbing the penetrating energy of an ultrashort pulsed laser escape from the region of material penetrated by a directly incident ultrashort pulsed laser. The Coulomb explosion, therefore, occurs due to the repulse of the ionized material part where electrons escape from it. After the time of phonon transport is reached, the residual heat from the penetration of the ultrashort pulsed laser is used to ionize the diffused material. Hence, for most materials, regardless of metal or nonmetals, the initial penetration of an ultrashort pulsed laser is directly recognized as the stage of optical effects. It is also considered as the stage of thermal effects that heat starts to diffuse after the transport time of phonons is arrived.

The ultrashort pulsed laser processing for materials involved the transfer between optical and thermal energy as well as the thermal transport between electrons and phonons in materials [13,19,21,24–26]. Based on the measured data [24,25], the optical and thermal effects on the ultrashort pulsed laser ablation for materials are considered to be two ablation regime: One is that the ablation depth per pulse for low laser fluences can be expressed by the product of optical absorption length and the logarithm of the ratio of laser fluence to laser threshold fluence and the other is that the ablation depth per pulse for high laser fluences is the product of thermal diffusive length and the logarithm of the ratio of laser fluence to higher-threshold fluence. According to the experimental observation [24], the phenomena of these two ablation regimes exist as a qualitative discrepancy. The discrepancy is that the ablated depth per pulse for the ablation regime of low laser fluences is dependent on the optical penetration depth (first logarithmic regime); no trace of molten material can be found in this regime, which is recognized directly as the optical ablation. At the second logarithmic regime of higher fluences, the trace of molten material is observed. The ripples on the ablation surface melt away, then rapid evaporation occurs and finally, plasma is formed on a very short time scale [27], which induces a deeper and narrower depth of thermal ablation due to the material rapid evaporation and ultra-intensity laser power-induced multiphoton absorption after a very short time of melt. This means that the period of the liquid phase is very short when the solid material is heated by the ultrashort pulsed laser of ultrahigh intensity (the very small focal spot of the ultrashort pulsed laser) and ultrahigh power (the very short pulse duration of the ultrashort pulsed laser). The processes of phase change of a solid transferred to a liquid and a liquid transferred to a vapor occur within a very short time. Therefore, the molten liquid phase hardly appears due to the ultrashort time of occurrence. Only a trace of molten material is observed. Hence, the model considering the solid–vapor interface is proposed for the ultrashort pulsed laser ablation of materials [17]. The ablation for this second logarithmic regime of higher fluences is due to heat-induced evaporation, which is recognized as thermal ablation. The heat is the residual energy of the directly incident laser pulse energy after the optical ablation.

There is a deviation from the square root of time of laser-pulsed duration scaling of laser damage fluence for pulses below 10 ps in dielectric materials. This phenomenon occurs because of the transition from a thermally dominated damage mechanism to one dominated by plasma formation on a time scale that is too short for significant energy transfer to the lattice, in which the material removal is

accompanied by a qualitative change in the morphology of the interaction site and essentially no collateral damage [27].

## 2. Analysis

A simplified photo-thermal model of the ultrashort pulsed laser processing for nonmetal thin films can be described by the following equation without involving multiphoton absorption and plasma formation in order to focus directly on the effects of incident optical ablation and sequent thermal ablation:

$$c\frac{\partial T}{\partial t} = k\frac{\partial^2 T}{\partial z^2} + \frac{3Q}{\pi\delta\sigma^2}(1-R)e^{-\left(\frac{t}{\tau_0}\right)^2 - \frac{z}{\delta} - 3\left(\frac{r}{\sigma}\right)^2} \tag{1}$$

where the thickness of the thin film is taken against the $z$-direction, which is the same as the penetration direction of an ultrashort pulsed laser. The direction $r$, which is the radial direction of cylinder coordinates and is perpendicular to the $z$-direction, is assumed to be infinite. Therefore, the incident laser penetration and heat transfer are considered to be only in the $z$-direction, which is the propagation direction of the ultrashort pulsed laser. Thermal diffusion in the direction of $r$ is assumed to be negligible due to the ultrashort time, so that heat transfer hardly occurs in the $r$ direction. The optical source of an ultrashort pulse laser in the second term of the right-handed side of Equation (1) can be expressed by a function of variables $r, z, t$. The $r$ coordinates indicate the direction perpendicular to the $z$-direction, where thermal diffusion is independent of the $r$ direction. Therefore, $r$ can be treated as constants or fixed values. These assumptions are applicable to the practical physical phenomenon for an ultrashort pulsed laser processing of materials, and hence, the model can be simplified to sharply show the effects of thermal and optical ablations. Due to the difference that occurs from nonmetals, the energy of an ultrashort pulsed laser first is absorbed by electrons of the electron sea in metals, and then the heat in electrons is transferred into the lattice after thermal diffusion time (phonon transport time). The two-temperature model is employed in the cases of the ultrashort pulsed laser processing of metals. The time for laser energy transferred to electrons is about of the order of $10^{-13}$ s. However, the time of the thermal diffusion is about of the order $10^{-12}$ s [28].

The other parameters in Equation (1) are summarized as follows: $T$ is the temperature in the workpiece. $k$ represents thermal conductivity. $c$ denotes volume heat capacity. $r$ and $z$ signify spatial coordinates. The symbol $t$ stands for time. A 185 fs pulse laser with a wavelength of 790 nm is used to conduct this laser ablation of ceramics, which mode is $TEM_{00}$ with the profile of Gaussian distribution. $Q$, $R$, $\delta$ and $\sigma$ correspondingly represent the laser power, reflectivity, optical absorption length and energy-distribution radius. The value 3 in Equation (1) are taken to ensure 90% of laser energy is included within the energy-distribution radius. $\tau_0$ symbolizes the pulse duration at FWHM.

The initial condition and boundary conditions are taken as:

$$T(z,0) = T_\infty,\ T(\infty,t) = T_\infty,\ -k\frac{\partial T(0,t)}{\partial z} + h[T(0,t) - T_\infty] = 0,\ T(f,t) = T_d \tag{2}$$

The evaporation ablation in this study is formulated by the energy balance at solid–vapor or decomposition interface:

$$-k\frac{\partial T}{\partial z} = \rho\gamma\frac{\partial f}{\partial t} \tag{3}$$

where $\gamma$ is the thermal activation energy or latent heat, which is related to bond energy of polymers or evaporation, respectively. Although an ultrashort pulsed laser ablation is a high nonequilibrium state, it can be approached using a quasi-equilibrium state in this study.

The non-dimensional parameters are defined as:

$$\theta = \frac{T - T_\infty}{T_d - T_\infty};\ \tau = \frac{t}{\tau_0};\ \xi = \frac{z}{L};\ \tau 0L = \frac{\tau_0}{\frac{cL^2}{k}} = \frac{\tau_0}{\tau L};\ H = \frac{hL}{k};\ R = \frac{r}{L};\ F = \frac{f}{L} \tag{4}$$

where $L$ is the thickness of the workpiece, $f$ represents the solid–vapor or decomposition interface, and $T_d$ is the evaporation temperature of materials or decomposition temperature of the polymer. Therefore, the non-dimensional governing equation yields:

$$\frac{\partial \theta}{\partial \tau} = \tau 0 L \times \frac{\partial^2 \theta}{\partial \xi^2} + \frac{J}{\delta c (T_d - T_\infty)}(1 - R)e^{-\tau^2 - \frac{L\xi}{\delta} - 3(\frac{LR}{\sigma})^2} \tag{5}$$

where the laser fluence $J$ equals to $3Q\tau_0/(\pi\sigma^2)$, and the non-dimensional form of the photo-thermal model for an ultrashort pulsed laser processing is shown in Equation (5). The coefficient of thermal diffusion term in the right-handed side of Equation (5) is $\tau 0 L$, which represents the ratio of the ultrashort laser pulse duration of thermal diffusion time. For femtosecond pulsed laser, the laser pulse duration is generally on the order of $10^{-15}$ s. The thermal diffusion time, $cL^2/k$, is generally on the order of $10^{-12}$ s for materials. The ratio of the ultrashort laser pulse duration of thermal diffusion time is generally far smaller than one. In this kind of case, the thermal diffusion term of the right-handed side in Equation (5) can be negligible because the order of other non-dimensional terms is near one.

The nondimensional initial and boundary conditions are:

$$\theta(\xi, 0) = 0, T(\infty, \tau) = 0, -\frac{\partial \theta(0, \tau)}{\partial \xi} + H\theta(0, \tau) = 0, T(F, \tau) = 1 \tag{6}$$

The non-dimensional formula of energy balance at the solid–vapor or decomposition interface becomes:

$$-\frac{\partial \theta}{\partial \xi} = \Omega_2 \frac{\partial F}{\partial \tau} \tag{7}$$

where:

$$\Omega_2 = \frac{\rho \gamma L^2}{k(T_d - T_\infty)\tau_0} \tag{8}$$

In order to analytically understand the effects of direct laser optical absorption and thermal diffusion on the temperature and the ablated depth, the Laplace transform is utilized to obtain the solution. If the Laplace transform of the temperature $\theta$ is symbolized by $\Theta$, the governing equation in the domain is:

$$s\Theta = \tau 0 L \times \frac{d^2 \Theta}{d\xi^2} + \frac{J}{\delta c (T_e - T_\infty)}(1 - R)e^{-\frac{L\xi}{\delta} - 3(\frac{LR}{\sigma})^2} \frac{\sqrt{\pi}}{2} e^{\frac{s^2}{4}} erfc(\frac{s}{2}) \tag{9}$$

and in the Laplace domain, the non-dimensional initial and boundary conditions are:

$$\Theta(\xi, 0) = 0, \ \Theta(\infty, s) = 0, -\frac{\partial \Theta(0, s)}{\partial \xi} + H\Theta(0, s) = 0 \ \Theta(\Gamma, s) = \frac{1}{s} \tag{10}$$

In the Laplace domain, the general solution to the initial condition is:

$$\Theta = c1 \times e^{-\sqrt{\frac{s}{\tau 0 L}}\zeta} + c2 \times e^{\sqrt{\frac{s}{\tau 0 L}}\zeta} - \frac{\frac{J}{\delta c (T_e - T_\infty)}(1 - R)e^{-\frac{L\xi}{\delta} - 3(\frac{LR}{\sigma})^2} \frac{\sqrt{\pi}}{2} e^{\frac{s^2}{4}} erfc(\frac{s}{2})}{(\tau 0 L \times \frac{L^2}{\delta^2} - s)} \tag{11}$$

The constant $c2$ is taken to be zero because the temperature is finite at $\xi \to \infty$. Using the boundary condition of the workpiece surface, we get the constant $c1$:

$$c1 = \frac{(-H + \frac{L}{\delta})}{(H - \sqrt{\frac{s}{\tau 0 L}})} \times \frac{\frac{-J}{\delta c (T_e - T_\infty)}(1 - R)e^{-3(\frac{LR}{\sigma})^2} \frac{\sqrt{\pi}}{2} e^{\frac{s^2}{4}} erfc(\frac{s}{2})}{(\tau 0 L \times \frac{L^2}{\delta^2} - s)} \tag{12}$$

Therefore,

$$\Theta = \Omega_1 \frac{e^{\frac{s^2}{4}} erfc(\frac{s}{2})}{(\tau 0 L \times \frac{L^2}{\delta^2} - s)} \times (\frac{H - \frac{L}{\delta}}{H - \sqrt{\frac{s}{\tau 0 L}}} e^{-\sqrt{\frac{s}{\tau 0 L}}\xi} - e^{-\frac{L\xi}{\delta}})$$

(13)

where:

$$\Omega_1 = \frac{J(1 - R)\sqrt{\pi} e^{-3(\frac{LR}{\sigma})^2}}{2\delta c(T_d - T_\infty)}$$

(14)

The contribution of thermal diffusion to the temperature is for the first term in the right-handed side of Equation (13). However, the contribution of direct laser optical absorption to the temperature is from the second term in the right-handed side of Equation (13).

The non-dimensional temperature in the time domain is derived from the inversion of the Laplace transform.

$$\theta = \Omega_1 \{ \sqrt{\tau_{0L}}(H - \frac{L}{\delta}) e^{(\frac{\tau_{0L}L^2}{2\delta^2})^2} \int_0^\tau [\frac{1}{\sqrt{\pi \tau}} e^{-\frac{\xi^2}{4u\tau_{0L}}} + (\sqrt{\tau_{0L}}H) e^{-H\xi} e^{\tau_{0L}H^2 u}] e^{\frac{\tau_{0L}L^2(\tau-u)}{\delta^2}}$$

$$[erf(\tau - u + \frac{\tau_{0L}L^2}{2\delta^2}) - erf(\frac{\tau_{0L}L^2}{2\delta^2})] du + e^{(\frac{\tau_{0L}L^2}{2\delta^2})^2 + \frac{\tau_{0L}L^2\tau}{\delta^2}} e^{-\frac{L\xi}{\delta}} [erf(\tau + \frac{\tau_{0L}L^2}{2\delta^2}) - erf(\frac{\tau_{0L}L^2}{2\delta^2})]\}$$

(15)

Substituting Equation (15) into the energy balance equation at the solid–vapor or decomposition interface, we get the equation for the solid–vapor or decomposition interface in the non-dimensional time domain.

$$\Omega_1 \{ \sqrt{\tau_{0L}}(H - \frac{L}{\delta}) e^{(\frac{\tau_{0L}L^2}{2\delta^2})^2} \int_0^\tau [\frac{1}{\sqrt{\pi \tau}} (\frac{F}{2u\tau_{0L}}) e^{-\frac{F^2}{4u\tau_{0L}}} + (\sqrt{\tau_{0L}}H^2) e^{-HF} e^{\tau_{0L}H^2 u}] e^{\frac{\tau_{0L}L^2(\tau-u)}{\delta^2}}$$

$$[erf(\tau - u + \frac{\tau_{0L}L^2}{2\delta^2}) - erf(\frac{\tau_{0L}L^2}{2\delta^2})] du + e^{(\frac{\tau_{0L}L^2}{2\delta^2})^2 + \frac{\tau_{0L}L^2\tau}{\delta^2}} (\frac{L}{\delta}) e^{-\frac{LF}{\delta}} [erf(\tau + \frac{\tau_{0L}L^2}{2\delta^2}) - erf(\frac{\tau_{0L}L^2}{2\delta^2})]\} = \Omega_2 \frac{\partial F}{\partial \tau}$$

(16)

Let the symbol $\Gamma$ represent the Laplace transform of the non-dimensional decomposition interface $F$. The energy balance equation of the solid–vapor or decomposition interface in the Laplace domain is

$$-\frac{\partial \Theta}{\partial \xi} = \Omega_2 s \Gamma$$

(17)

Combining Equation (13) with Equation (17), one gets the equation for the solid–vapor or decomposition interface in the non-dimensional Laplace domain.

$$\Omega_1 \times \frac{e^{\frac{s^2}{4}} erfc(\frac{s}{2})}{(\tau 0 L \times \frac{L^2}{\delta^2} - s)} \times (\sqrt{\frac{s}{\tau 0 L}} \times \frac{H - \frac{L}{\delta}}{H - \sqrt{\frac{s}{\tau 0 L}}} e^{-\sqrt{\frac{s}{\tau 0 L}}\Gamma} - \frac{L}{\delta} e^{-\frac{L\Gamma}{\delta}}) = \Omega_2 s \Gamma$$

(18)

After taking the logarithm for Equation (18) and re-arranging the formula, we obtain the relationship between the laser fluence and the ablated depth in the Laplace domain

$$Log(J) = -Log[\frac{k \times \tau 0 \times (1 - R)\sqrt{\pi} e^{-3(\frac{r}{\sigma})^2} e^{\frac{s^2}{4}} erfc(\frac{s}{2})}{2\delta c \rho \gamma L^2 \times (\tau 0 L \times \frac{L^2}{\delta^2} - s)} \times (\sqrt{\frac{s}{\tau 0 L}} \times \frac{H - \frac{L}{\delta}}{H - \sqrt{\frac{s}{\tau 0 L}}} e^{-\sqrt{\frac{s}{\tau 0 L}}\Gamma} - \frac{L}{\delta} e^{-\frac{L\Gamma}{\delta}})]$$

$$+Log[s\Gamma]$$

(19)

The integral term at the left-handed side of Equation (16) is from the thermal diffusion. The other terms are attributed to the direct laser fluence. For polymers, the order of the thermal and optical properties is $T_d \sim 10^2\,°C, c \sim 10^6\,J \cdot m^{-3}\,°C^{-1}, L \sim 10^{-3}\,m, \tau_0 \sim 10^{-12}\,s, k \sim 10^0\,W \cdot m^{-1}\,°C^{-1}, h \sim 10^2\,W \cdot m^{-2}\,°C^{-1}, \rho \sim 10^3\,kg \cdot m^{-3}, \delta \sim 10^{-6}\,m$. Therefore, the order of the non-dimensional parameters is $\tau_{0L} \sim 10^{-12}$,

$L/\delta \sim 10^3$. These results give that the effect of direct laser fluence on the ablation rate is far larger than the thermal diffusion.

Finally, Equation (16) is simplified as

$$\Omega_1 \{e^{(\frac{\tau_{0L}L^2}{2\delta^2})^2 + \frac{\tau_{0L}L^2\tau}{\delta^2}}(\frac{L}{\delta})e^{-\frac{LF}{\delta}}[erf(\tau + \frac{\tau_{0L}L^2}{2\delta^2}) - erf(\frac{\tau_{0L}L^2}{2\delta^2})]\} = \Omega_2 \frac{\partial F}{\partial \tau} \tag{20}$$

Integrating Equation (20), we get

$$\frac{-\frac{J(1-R)\sqrt{\pi}e^{-3(\frac{LR}{\delta})^2}}{2\delta c(T_d-T_\infty)}}{\Omega_2}\int_{\tau_e}^{\tau}\{e^{(\frac{\tau_{0L}L^2}{2\delta^2})^2 + \frac{\tau_{0L}L^2\tau}{\delta^2}}[erf(\tau + \frac{\tau_{0L}L^2}{2\delta^2}) - erf(\frac{\tau_{0L}L^2}{2\delta^2})]\}d\tau + 1 = e^{-\frac{LF}{\delta}} \tag{21}$$

After taking the logarithm of two sides of Equation (21), the relationship between the laser fluence and the ablation depth is obtained.

$$Log_e J + Log_e(\frac{-\frac{(1-R)\sqrt{\pi}e^{-3(\frac{LR}{\delta})^2}}{2\delta c(T_d-T_\infty)}}{\Omega_2}\int_{\tau_e}^{\tau}\{e^{(\frac{\tau_{0L}L^2}{2\delta^2})^2 + \frac{\tau_{0L}L^2\tau}{\delta^2}}[erf(\tau + \frac{\tau_{0L}L^2}{2\delta^2}) - erf(\frac{\tau_{0L}L^2}{2\delta^2})]\}d\tau + 1) = \frac{LF}{\delta} \tag{22}$$

From Equation (22), it is found that the ablation depth is proportional to the logarithm of the laser fluence by a ratio factor $\delta/L$, which is the optical absorption length. This follows Beer's model.

The Reimann-sum approximation was used to calculate the inversion of the Laplace transform.

$$\theta(\tau, \xi) = \frac{e^{\varepsilon\tau}}{\tau}\{\frac{\Theta(\varepsilon, \xi)}{2} + \text{Re}[\sum_{n=1}^{N}\Theta(\varepsilon + \frac{in\pi}{\tau}, \xi)(-1)^n]\} \tag{23}$$

## 3. Results and Discussion

The ablated depth of an ultrashort pulsed laser was generally recognized as two mechanisms. One is optical ablation, and the other is thermal ablation. Optical ablated depth for an ultrashort pulsed laser can be estimated by the product of optical absorption length and the logarithm of the ratio of laser fluence to laser fluence threshold ($D_o = l_o \text{Log}(\phi/\phi_{oth})$ [13,24,25]. Thermally ablated depth for an ultrashort pulsed laser can be predicted from the product of thermal diffusion length and the logarithm of the ratio of laser fluence to laser fluence threshold ($D_d = l_d \log_{10}(\phi/\phi_{dth})$ [19,24]. This work elucidates the effects of thermal ablation and optical ablation on an ultrashort pulsed laser processing for materials using the Laplace transform method. The Laplace transform method can divide the solutions of an analytical model of an ultrashort pulsed laser logarithm processing for material into two parts. One is optical ablation, and the other is thermal ablation. Therefore, the optical and thermal ablated characteristics can be, respectively, analyzed and discussed. In this study, the analysis of an ultrashort pulsed laser processing of polymers and ceramics are shown in the following figures, where the data are calculated by the Equation (19) in the ANALYSIS of Section 2.

Figure 1a shows that an ultrashort pulsed laser ablation depth for poly(methyl methacrylate) (PMMA) was predicted by this work and measured by published papers [13], respectively. The predicted ablation depth per pulse versus laser fluence agrees with the measured ablation depth per pulse versus laser fluence. The horizontal axis in Figure 1a is in the logarithm scale, which is different from the horizontal axis of the linear scale in Figure 1b. In Figure 1a, with the horizontal axis of the logarithm scale, the ablated depth per pulse increases with the increasing laser fluence and the rate of increase of the ablated depth per pulse with laser fluence seems to be larger for high laser fluence than that for low laser fluence. However, the increase of ablated depth per pulse with laser fluence is almost equal to that for high laser fluence in Figure 1b, where the horizontal *x*-axis for laser fluence is in the linear scale. This indicates that the obvious increase of the ablated depth per pulse with laser fluence for the high laser fluence is not real in Figure 1a, in which the horizontal axis of laser fluence is

plotted in the scale. Nevertheless, the ablated depth per pulse of the ultrashort pulsed laser processing for PMMA is still increasing with the increasing laser fluence in Figure 1a,b. Based on the Laplace transform, the optical and thermal ablation can be analyzed, respectively. The two straight lines approximating the measured data are also plotted in Figure 1a with the horizontal axis of the logarithm scale. The straight line at low laser fluence is considered to be directly optical ablation, which is linearly proportional to the optical absorption length and follows logarithmic Beer's law (Bremsstrahlung's absorption) ($D_o = l_o \mathrm{Log}(\phi/\phi_{oth})$). On the other hand, the straight line at high laser fluence is recognized as the sequentially thermal ablation, which is linearly proportional to thermal diffusion length and follows the relation ($D_d = l_d \log_{10}(\phi/\phi_{dth})$). In Figure 1, the threshold laser fluences of $\phi_{oth}$ and $\phi_{dth}$ are, respectively, about 8000 J/m$^2$ and 48,000 J/m$^2$. From Equation (22), it is also found that the ablation depth per pulse is linearly proportional to the logarithm of the laser fluence by a factor of the optical absorption length.

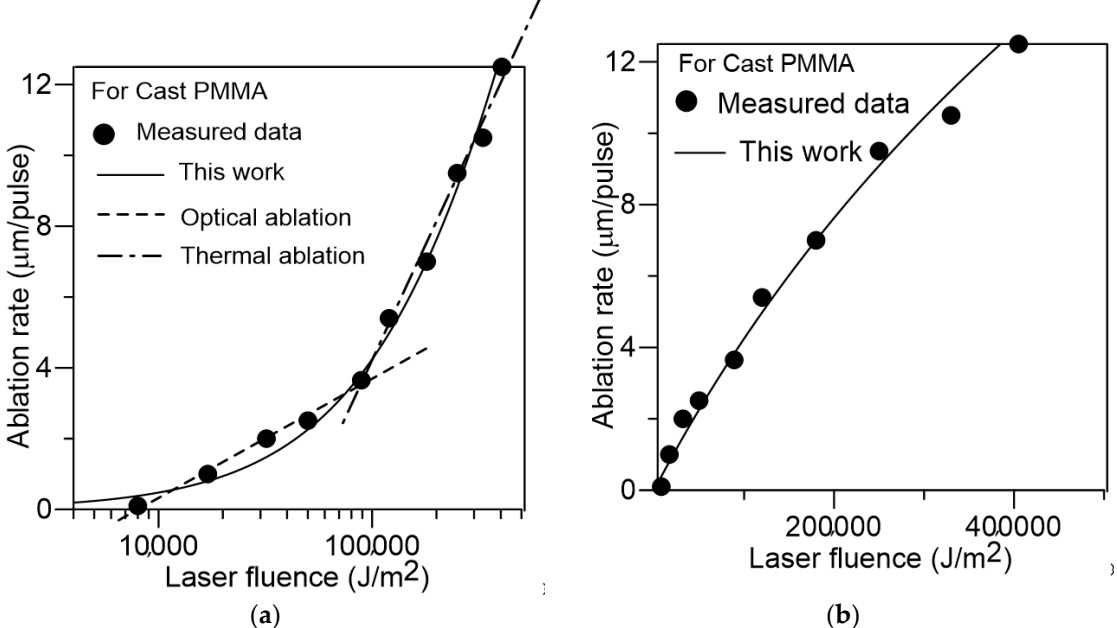

**Figure 1.** Ablation rate versus laser fluence of poly(methyl methacrylate) (PMMA) for the solid line predicted by this work and the circle symbols measured by the published paper [13] (**a**) with the horizontal axis of the logarithm scale, and (**b**) with the horizontal axis of the linear scale.

Figure 2a shows that the ablated depth per pulse varies with the logarithm of laser fluence for acrylonitrile butadiene styrene/polyvinyl chloride (ABS/PVC), the ablated depth per pulse versus the logarithm of laser fluence is predicted from this work, the ablated depth per pulse versus the logarithm of laser fluence is measured by the published paper [13], the ablation is only performed by direct optical energy of laser for low laser fluence, and the depth per pulse is ablated by initial optical energy as well as sequent thermal diffusion for high laser fluence. The relation of the measured ablation depth per pulse to the logarithm of laser fluence is consistent with the results calculated by this work for laser fluence <200,000 J/m$^2$ but deviates from the calculated results for laser fluence >200,000 J/m$^2$. The possible reason could be as results of measured errors. The two short lines for low and high laser fluence, respectively, represent the optical and thermal ablations, which are plotted according to the measured data and this was in agreement with Beer's exponential law with the relations of ($D_o = l_o \mathrm{Log}(\phi/\phi_{oth})$) and ($D_d = l_d \log_{10}(\phi/\phi_{dth})$) where the threshold laser fluences of $\phi_{oth}$ and $\phi_{dth}$ are about 6000 J/m$^2$ and 30,000 J/m$^2$, respectively. In Figure 2b, the increase of the ablated depth per pulse with laser fluence is smaller for high laser fluence than that for low laser fluence in the linear scale of the horizontal axis, which is different from the horizontal axis of laser fluence in the logarithm scale for

Figure [2]a. The increase of ablated depth per pulse with the logarithm of laser fluence seems to be far larger for high laser fluence than that for low laser fluence in Figure [2]a.

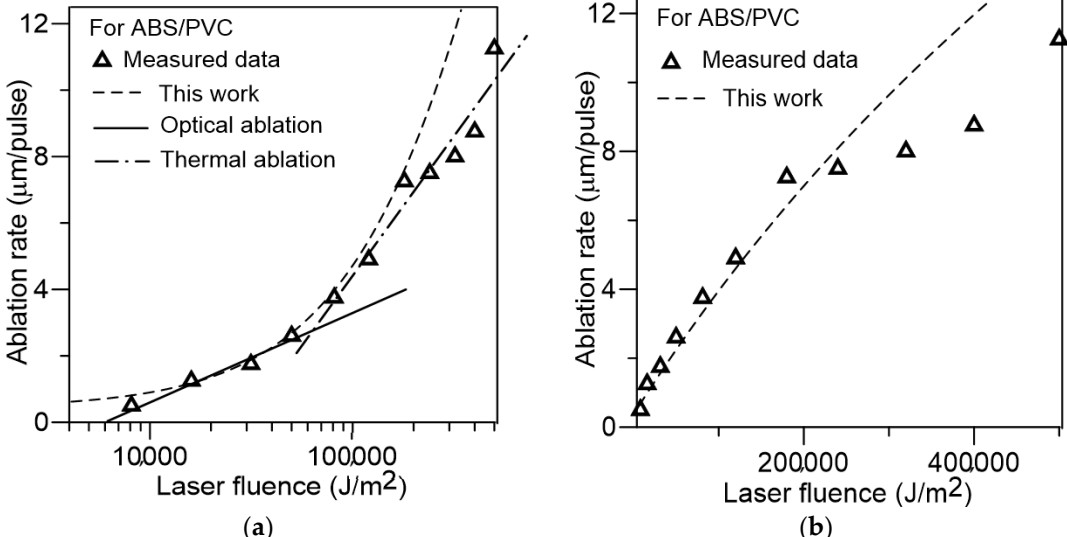

**Figure 2.** Ablation rate versus laser fluence of acrylonitrile butadiene styrene/polyvinyl chloride (ABS/PVC) for the dashed line predicted by this work and the triangle symbols measured by the published paper [13] (**a**) with the horizontal axis of the logarithm scale, and (**b**) with the horizontal axis of the linear scale.

For $Al_2O_3$ ceramics, the results obtain from this research work, the data measured from published paper [29], optical and thermal ablation of the ablated depth per pulse versus laser fluence are plotted in Figure [3]a, where the horizontal axis is the logarithm of laser fluence. The ablated depth per pulse versus laser fluence predicted by this work is consistent with that measured from the published paper [29]. The two short straight lines at the low and high laser, respectively, represent the optical and thermal ablation based on the measured data, which follow the two relations, $D_o = l_o Log(\phi/\phi_{oth})$ and $D_d = l_d \log_{10}(\phi/\phi_{dth})$. The $\phi_{oth}$ and $\phi_{dth}$ are the thresholds for optical ablation and thermal ablation, respectively. The value of $\phi_{oth}$ is the intersection of the horizontal axis in the logarithm-scaled laser fluence and the short straight line at low laser fluence. The $\phi_{dth}$ is the value of the intersection of the horizontal axis in the logarithm-scaled laser fluence and the short straight line at high laser fluence. Figure [3]a indicates that the threshold laser fluences of $\phi_{oth}$ and $\phi_{dth}$ are about 10,200 J/m$^2$ and 103,000 J/m$^2$, respectively. Similar to Figures [1] and [2], Figure [3]a,b also reveal that the ablated depth per pulse increases with the increasing laser fluence. Although it has also shown that the increase in the ablated depth per pulse with the logarithm of laser fluence is faster for high laser fluence than that for low laser fluence in Figure [3]a, the increase of the ablated depth per pulse with laser fluence in the linear scale of the horizontal axis is obviously slower for high laser fluence than that for low laser fluence in Figure [3]b. This indicates that the increase of the ablated depth per pulse with laser fluence is larger due to optical ablation than that due to thermal ablation in Figure [3]b, unlike Figure [3]a. This is because the optical energy of the ultrashort pulsed laser is the main heat source, which induces the optical ablation during the ultrashort pulsed duration of the laser. However, the thermal ablation depends on the thermal diffusion of the residual heat of the optical energy of the ultrashort pulsed laser after the transport time of phonons in the lattice is arrived.

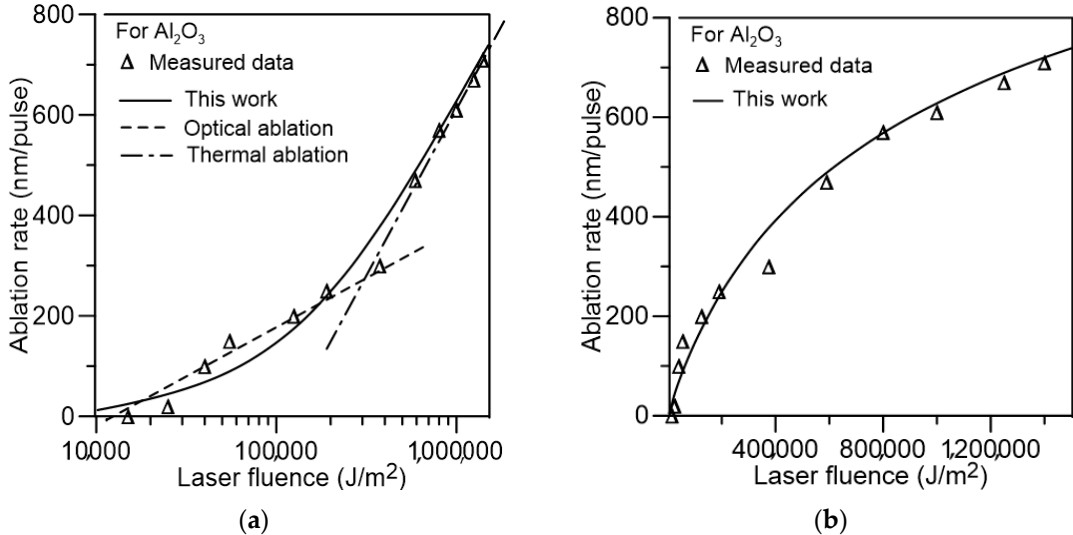

(**a**)              (**b**)

**Figure 3.** Ablation rate versus laser fluence of $Al_2O_3$ ceramics for the solid line predicted by this work and the triangle symbols measured by the published paper [29] (**a**) with the horizontal axis of the logarithm scale, and (**b**) with the horizontal axis of the linear scale.

The ablated depths per pulse of an ultrashort pulsed laser processing for Au calculated by this work and measured by the published paper [25] are plotted in Figure 4 for different fluences. The ablated depths per pulse of Au calculated by this work agree with the measured data for laser fluence <6500 $J/m^2$ but deviate from the measured data for laser fluence >6500 $J/m^2$. Moreover, the variation of the ablated depth per pulse of Au with the logarithm of laser fluence is also plotted in Figure 4 by ignoring the effect of thermal ablation, which is calculated by this work and shown as a dashed line in Figure 4. The dashed line approximates a straight line and represents the optical ablation without considering the thermal diffusion. This reveals that the model proposed by this work can validate the two ablated mechanisms, such as optical and thermal ablation of the ultrashort pulsed laser processing of materials.

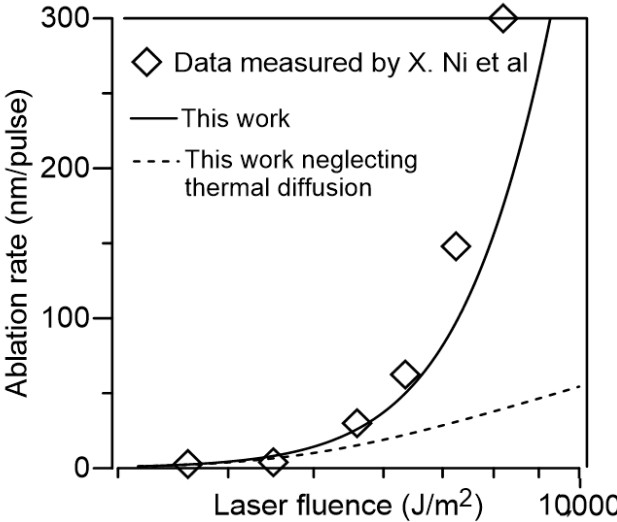

**Figure 4.** Variation of ablation rate with laser fluence for the solid line calculated by this work, the diamond symbols measured by the published paper [25], and the dashed line predicted by this work without considering thermal diffusion.

The moving rate of the solid–vapor interface for optical and thermal ablation is plotted in Figure 5 during the process of ablation. The optical energy of the ultrashort pulsed laser is only one heat source

in the process of ablation, which triggers the ablation till the pulse ends. Then, heat is transferred into the lattice through thermal diffusion after phonon transport time has arrived. When the residual heat directly from the incident optical energy of the ultrashort pulsed laser is thermally diffused into the lattice and can raise or maintain the temperature of materials above the evaporated temperature, thermal ablation occurs. Therefore, the optical ablation occurs during the period of the laser pulse. This is the reason why the increase of solid–vapor interface with time for optical ablation initially is high and then rapidly descends in Figure 5. In sequence, if the residual heat diffused into the lattice can maintain the temperature above the evaporated temperature, the thermal ablation appears after thermal diffusion time. Therefore, the thermal ablation is larger for the laser fluence = 100,000 J/m$^2$ than that for laser fluence = 10,000 J/m$^2$ because the heat transferred into the lattice for high laser fluence is much more than that for low laser fluence after the duration of a laser pulse, and this can maintain the temperature above the evaporated temperature. Bold line and bold dashed line, respectively, represent the moving velocity of solid–vapor interface for optical and thermal ablation at laser fluence = 100,000 J/m$^2$, as shown in Figure 5. The fine line and fine dashed line, respectively, represent the moving velocity of solid–vapor interface for optical and thermal ablation at laser fluence = 10,000 J/m$^2$ in Figure 5. Evidently, the high laser fluence gives the fast-moving velocity of the solid–vapor interface of optical and thermal ablation in Figure 5. Therefore, while the increase of the ablated depth with laser fluence is smaller for the high laser fluence than that for the low laser fluence because the thermal ablation only uses the residual heat of the ultrashort pulsed laser, the deep depth of ablation is still obtained for high laser fluence.

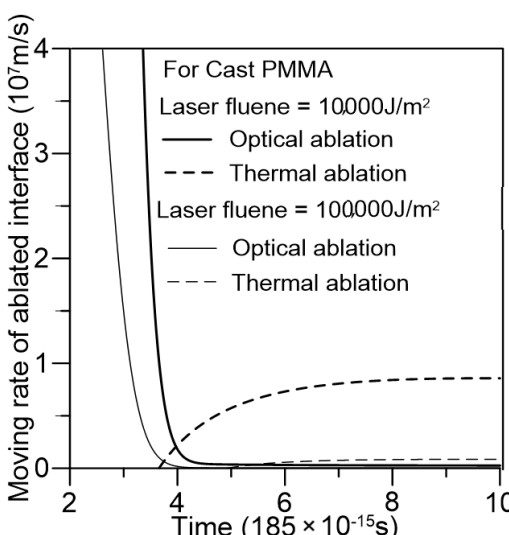

**Figure 5.** Moving rate of ablated interface versus time of optical ablation and thermal ablation for the bold line and bold dashed line with laser fluence 100,000 J/m$^2$ as well as the fine line and fine dashed line with laser fluence 10,000 J/m$^2$.

## 4. Conclusions

- The model proposed by this work can, respectively, divide the analysis of an ultrashort pulsed laser processing of materials into the optical and thermal effects using the Laplace transform;
- The ablated depth per pulse versus laser fluence predicted by this study agrees with the data measured by the published paper. Both optical and thermal ablations follow Beer's exponential law based on optical absorption length and thermal diffusion length, respectively;
- It has also been shown that optical and thermal ablation, respectively, occur in low and high laser fluences because the optical ablation depends on the only main source of the direct incident ultrashort pulsed laser. However, the thermal ablation is governed by the residual heat of the directly incident ultrashort pulsed laser after the optical ablation. Therefore, the thermal ablation

is more obvious for high laser fluences. If the laser fluence is not high enough, the thermal ablation is insignificant;

- This paper analytically indicates that the depth per pulse of optical ablation approximates the product of optical absorption length and the logarithm of the ratio of laser fluence to the laser fluence threshold. The variation of ablation rate with laser fluence for data calculated by this work and measured by the published paper is shown in the coordinates with the horizontal axis of the logarithm scale and linear scale. In the coordinates with the horizontal axis of the logarithm scale, the rate of increase of the ablated depth per pulse with laser fluence is faster at higher laser fluences than that at low laser fluences. However, the rate of increase of the ablated depth per pulse with laser fluence is actually lower at high laser fluences than that at low laser fluences in the coordinates with the horizontal axis of linear scale because the thermal ablation using the residual heat of the directly incident ultrashort pulsed laser governs at high laser fluences.

**Author Contributions:** Conceptualization, C.-Y.H. and D.-K.Q.; methodology, C.-Y.H. and C.-W.X.; software, C.-W.X.; validation, C.-Y.H., D.-K.Q. and C.-W.X.; formal analysis, C.-Y.H.; investigation, C.-Y.H.; resources, C.-W.X.; data curation, C.-W.X.; writing–original draft preparation, C.-W.X.; writing–review and editing, C.-Y.H.; visualization, C.-W.X.; supervision, C.-Y.H.; project administration, C.-W.X.; funding acquisition, D.-K.Q. All authors have read and agreed to the published version of the manuscript.

**Funding:** The support by Dongguan Social Science and Technology Development (Major) Project (Grants No. 2020507156175), the Natural Science Foundation of Guangdong province, China (2018A0303070004), Research Item of Natural Science of Guangdong University of Petrochemical Technology (2019rc66), the Ministry of Science and Technology under grant no. MOST 108–2221-E-146–003-, Guangdong Educational Department of Scientific Research Project under grant No. 2017GKTSCX105, Dongguan Polytechnic Scientific Research Fund through grant No. 2017a04 and Dongguan Polytechnic Quality Engineering Project under No. JGZD201826 is gratefully acknowledged.

**Acknowledgments:** Thank Bor-Chyuan Chen and Chia-Chieh Ho for giving valuable opinions on this paper.

**Conflicts of Interest:** The authors declare no conflict of interest.

## Abbreviations

PMMA　　　Poly(methyl methacrylate)
ABS　　　　Acrylonitrile butadiene styrene
PVC　　　　Polyvinyl chloride

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
