# Peer review of "Analysis of Direct Optical Ablation and Sequent Thermal Ablation for the Ultrashort Pulsed Laser Photo-Thermal Micromachining"

_coatings, doi:10.3390/coatings10121151_

Round 1

Reviewer 1 Report

In the manuscript, the authors develop a simple theoretical model to describe the ablation induced by a ~100fs duration laser pulse on a solid material. A solution is found by means of Laplace transform, providing an expression where contributions attributed by the authors to so-called optical and thermal ablation can be identified. Predictions by this model are then compared to the ablation rate as reported in previous literature.

In my opinion, the manuscript may be interesting from a practical point of view, and it may deserve publication. However, I have to say that the quality of the english language used is very very very poor, and quite often prevents the reader from following the arguments or requires a rather huge effort to do so. I thus feel obliged to suggest a major revision, since without a profound revision of the english language any reader would be discouraged from read the paper.

As for the scientific content, I have only minor comments.

  • Please, clarify why a (Dirichlet) boundary condition is used, in Eq. 2, for T at +infinity, although the thickness of the target is supposed finite.
  • Please, check the x range/label in the plot of Fig. 5.

Author Response

Responses to the comments of reviewers:

Reviewer #1

Comment #1 of reviewer #1:  I thus feel obliged to suggest a major revision, since without a profound revision of the english language any reader would be discouraged from read the paper.

Response to comment #1 of reviewer #1: Thank reviewer for valuable comments. The English has been edited by the professional company of proofreading and editing services. Please see the revised manuscript of English proofreading. The certification of English proofreading is attached in the last part.  

Comment #2 of reviewer #1: Please, clarify why a (Dirichlet) boundary condition is used, in Eq. 2, for T at +infinity, although the thickness of the target is supposed finite.

Response to comment #2 of reviewer #1: Thank reviewer for valuable comments.   

An ultrashort pulse laser micromachining for non-metal materials with a general- scaled size is nvestigated in this paper. The thicknesses of workpieces are about 1000~100000 μm in this study. However, the ablated depth per pulse of an ultrashort pulse laser micromachining for materials is about 0.1~10μm in this study. Therefore, the thickness of materilas is far larger than the depth per pulse ablated by an ultrashort pulse laser in this study. The bottom of workpiece is recognized to be very far distance away from the heat-affected zone of laser penetration and its temperature approaches room temperature (or the ambient temperature). Consequently, the boundary condition at the bottom of workpiece is assumed to be infinity in the dirction of z and its temperature is equal to room temperature (or the ambient temperature).     

Comment #3 of reviewer #1: Please, check the x range/label in the plot of Fig. 5.

Response to comment #3 of reviewer #1: The symbol in the horizontal axis of Fig. 5 occurs error in the transfer process from graphical software to WORD software.  The symbol, ×, between 185 and 10-15s represents the product of 185 and 10-15s. The label in the horizontal axis of Fig. 5 has been modified. Please refer to the revised manuscript.

Reviewer 2 Report

The manuscript in question presents novelty and is interesting for scientific community. It's describe the laser ablation description for samples which are non-metals ; so in consequence it does not appeal to Two Temperature Model.

The results are original and correct in my opinion.

I have to make one short criticism: because Laplace transform is used extensively it would be better to give the source of inspiration about it: like for example to put some references (like mathematics  books or articles or specialized software in Laplace transform in MATHEMATICA. ) I suggest that a mathematical annex it would be very good in order to explain how do you calculate the Laplace transform. 

In rest I do not have other observations; and with this small suggestion I sustain the paper to be published.

Author Response

Responses to the comments of reviewers:

Reviewer #2

Comment #1 of reviewer #2: I have to make one short criticism: because Laplace transform is used extensively it would be better to give the source of inspiration about it: like for example to put some references (like mathematics  books or articles or specialized software in Laplace transform in MATHEMATICA. ) I suggest that a mathematical annex it would be very good in order to explain how do you calculate the Laplace transform.

Response to comment #1 of reviewer #2: The Laplace transform technique can be employed to analyze the phase-change problems. (Ku, J. Y.; Chan, S. H. A generalized Laplace transform technique for phase-change problems. ASME Journal of Heat Transfer 1990 , 112, 495-496).

This study used the software “MATHEMATICA 10.0” to calculate the inversion of Laplace transform and equations in the “Analysis” section of this manucsript.

The Reimann-sum approximation was used to calculate the inversion of Laplace transform.

Equation (Attached please see PDF file)

where Re denotes the real part of the complex function. For the effective convergence, the value of ε × τ is generally chosen to be 1.3.

Reviewer 3 Report

I can help but wonder why 2 authors are from medicine world. Please define the link with medicine doctors.

Department of Chinese Medicine, Buddhist Dalin Tzu Chi General Hospita; csg3411@yahoo.com.tw 11

Department of Emergency Medicine, Mackay Memorial Hospital; 2563794452@qq.com

Really, I do not see the link. Perhaps it is my error, but it sound very weird. PLEASE EXPLAIN IT.

References: Please include the models by Analysis of the regimes in the scanner-based laser in Optics and Lasers in engineering 90, 72-80 because some ideas can be linked with the work.

Micomachining is key in several applications: most of the references are from China. CVheck Ukar, or Lamikiz and other references about laser engraving, ablation, etc.

What are doing people of Medicine as authors in this paper. We must be very careful here. I do not see any link with traditional chinese medicine and laser abalation.

Laser ablation has direct connection with laser polishing. Ukar published good ideas about that, specially Roughness prediction on laser polished surfaces, Journal of Materials Processing Technology 212 (6), 1305-1313 with models of application for your approach. ukar and others worked on your fields.

Your references are all Chinese?

Results are nice.

Author Response

Responses to the comments of reviewers:

Reviewer #3

Comment #1 of reviewer #3: Department of Chinese Medicine, Buddhist Dalin Tzu Chi General Hospita; csg3411@yahoo.com.tw; Department of Emergency Medicine, Mackay Memorial Hospital; 2563794452@qq.com

Really, I do not see the link. Perhaps it is my error, but it sound very weird. PLEASE EXPLAIN IT.

Response to comment #1 of reviewer #3: Contributions of Bor-Chyuan Chen and Chia-Chieh Ho to this manuscript are only a little. Instead, they are listed in the “Acknowledgement” of this manuscript. Please see the revised manuscript. 

Comment #2 of reviewer #3: Please include the models by Analysis of the regimes in the scanner-based laser in Optics and Lasers in engineering 90, 72-80 because some ideas can be linked with the work.

Response to comment #2 of reviewer #3: Thank reviewer for your kind suggestions. The paper has been cited in this manuscript. Please see the reference [1] of the revised manuscript.  

Comment #3 of reviewer #3: Micomachining is key in several applications: most of the references are from China. CVheck Ukar, or Lamikiz and other references about laser engraving, ablation, etc.

Response to comment #3 of reviewer #3: All references cited by this manuscript are from the international journals (not Chinese journal or journals issued in China), in which papers are written in English. Other references about CVheck Ukar, or Lamikiz  have been cited in the “Introduction” of this manuscript. Please refer to the reference [1,2] of the revised manuscript. 

Comment #4 of reviewer #3: What are doing people of Medicine as authors in this paper. We must be very careful here. I do not see any link with traditional chinese medicine and laser abalation.

Response to comment #4 of reviewer #3: Contributions of Bor-Chyuan Chen and Chia-Chieh Ho to this manuscript are only a little. Instead, they are listed in the “Acknowledgement” of this manuscript. Please see the revised manuscript.  

Comment #5 of reviewer #3: Laser ablation has direct connection with laser polishing. Ukar published good ideas about that, specially Roughness prediction on laser polished surfaces, Journal of Materials Processing Technology 212 (6), 1305-1313 with models of application for your approach. ukar and others worked on your fields.

Response to comment #5 of reviewer #3: Good ideas about laser polishing were proposed by Ukar whose paper has been cited in the “Introduction” of this manuscript. Please refer to the reference [2] of the revised manuscript. 

Comment #6 of reviewer #3: Your references are all Chinese?

Response to comment #6 of reviewer #3: All references cited by this manuscript are from the international journals (not Chinese journal or journals issued in China), in which papers are written in English.

Comment #7 of reviewer #3: Results are nice.

Response to comment #7 of reviewer #3: Thank reviewer for comments.

Reviewer 4 Report

The paper can be published in its present form, but after editing the English. English is sometimes quite confusing in this work.

Author Response

Responses to the comments of reviewers:

Reviewer #4

Comment #1 of reviewer #4: The paper can be published in its present form, but after editing the English. English is sometimes quite confusing in this work.

Response to comment #1 of reviewer #4: Thank reviewer for valuable comments. The English has been edited by the professional company of Proof Reading and Editing Services. Please see the revised manuscript of English proofreading. The certification of English proofreading is attached in the last part (please see the PDF file).

Round 2

Reviewer 3 Report

This is a much better version:

Line 17: there are 2 dots.

Line 39   My guess is that is drilling…not dilling

Can you rigth better the first paragraph of the work: you can separate firstly: include the paper about heat treatments ( refs 3,4..) later those about cutting, welding??, afterward the polishing ones, and finally the additive. This is more sutiable because the first applications were for heat treatment. Drilling by laser is key as well. I enjoyed some Works in Journal: https://journals.sagepub.com/home/pib the jorunal is very industrial focused. I am sure you can find more references in journals with authors from America or Europe.

Conclusions: could you give some of them as bullets, one per each highlight?

¿ do you have some image of treated  surfaces by microscopy?

Please tak your time to improve the paper. Last recommendation were accomplihed too ligthly.

Author Response

Response to reviewers’ comments of manuscript No.coatings-965425 

(Analysis of direct optical ablation and sequent thermal ablation for the ultrashort pulsed laser photo-thermal micromachining )

The authors would like to thank reviewers’ valuable comments to make this paper more completed and free of errors. Revisions were made in this paper, and the responses to reviewers’ comments are summarized in the following:

Responses to the comments of reviewers:

Reviewer #3

Comment #1 of reviewer #3:  Line 17: there are 2 dots.

Response to comment #1 of reviewer #3: Thank reviewer for reminders. 2 dots in line 17 have been corrected. Please see the revised manuscript.

Comment #2 of reviewer #3: Line 39   My guess is that is drilling…not dilling.

Response to comment #2 of reviewer #3: Thank reviewer for kind reminders. The “dilling” has been modified into the “drilling”.    

Comment #3 of reviewer #3: Can you rigth better the first paragraph of the work: you can separate firstly: include the paper about heat treatments ( refs 3,4..) later those about cutting, welding??, afterward the polishing ones, and finally the additive. This is more sutiable because the first applications were for heat treatment. Drilling by laser is key as well.

Response to comment #3 of reviewer #3: Thank reviewer for kind suggestions. The sequence of arrangement of the references has been modified according to the reviewer’s comment.

  1. Introduction

Laser can be applied in many machinings of materials such as laser heat treatments [1], drilling [2], cutting [3-5], welding [6-8], hardening process [9], polished surfaces [10], nanometer-scaled processing [11], etc. The nanometer-scaled processing was generally achieved by an ultrashort pulsed laser. An ultrashort pulsed laser has the characteristics of the general laser such as contactless processing, wear-free, clean surfaces, flexible arrangements of beam deflection and ......

Comment #4 of reviewer #3: I enjoyed some Works in Journal: https://journals.sagepub.com/home/pib the jorunal is very industrial focused. I am sure you can find more references in journals with authors from America or Europe.

Response to comment #4 of reviewer #3: Thank reviewer for kind suggestions. This manuscript cited some works in Journal: https://journals.sagepub.com/home/pib.

  1. Shiva, S.;Palani, I. A.; Paul, C. P.; Singh, B. Laser annealing of laser additive–manufactured Ni-Ti structures: An experimental–numerical investigation. Proceedings of the Institution of Mechanical Engineers, Part B: Journal of Engineering Manufacture 2018, 232, 1054-1067.(Idia)
  2. Negarestani, R.;Li L. Fibre laser cutting of carbon fibre–reinforced polymeric composites. Proceedings of the Institution of Mechanical Engineers, Part B: Journal of Engineering Manufacture 2013, 227, 1755-1766.(UK)
  3. Madić, ; Mladenović, S.; Gostimirović, M.; Radovanović, M.; Janković, P.Laser cutting optimization model with constraints: Maximization of material removal rate in CO2 laser cutting of mild steel. Proceedings of the Institution of Mechanical Engineers, Part B: Journal of Engineering Manufacture 2020, 234 ,1323-1332.(Serbia)
  4. Shukla, P. Investigation into the high-speed laser welding feasibility of tin-plated steels available for three-piece food packaging can manufacture. Proceedings of the Institution of Mechanical Engineers, Part B: Journal of Engineering Manufacture2014, 228, 715-729.(UK)

The cited papers with authors from America or Europe by this manuscript are as follows:

  1. Negarestani, R.; Li L. Fibre laser cutting of carbon fibre–reinforced polymeric composites. Proceedings of the Institution of Mechanical Engineers, Part B: Journal of Engineering Manufacture 2013, 227,1755-1766.(UK)
  2. Madić, ; Mladenović, S.; Gostimirović, M.; Radovanović, M.; Janković, P.Laser cutting optimization model with constraints: Maximization of material removal rate in CO2 laser cutting of mild steel. Proceedings of the Institution of Mechanical Engineers, Part B: Journal of Engineering Manufacture 2020, 234 ,1323-1332.(Serbia)
  3. Shukla, P. Investigation into the high-speed laser welding feasibility of tin-plated steels available for three-piece food packaging can manufacture. Proceedings of the Institution of Mechanical Engineers, Part B: Journal of Engineering Manufacture2014, 228, 715-729.(UK)
  4. Martínez,; Lamikiz, A.; Ukar, E.; Calleja, A.; Arrizubieta, J. A.; Lopez de Lacalle, L. N. Analysis of the regimes in the scanner-based laser hardening process. Optics and Lasers in Engineering 2017, 90, 72-80.(Spain)
  5. Ukar, ; Lamikiz, A.; Martínez, S.; Tabernero, I.; López deLacalle, L.N. Roughness prediction on laser polished surfaces. Journal of Materials Processing Technology 2012, 212, 1305-1313.( Spain)
  6. Garcia-Lechuga, M.; Gebrayel El Reaidy G.; Ning H.; Delaporte, P.; Grojo, D.Assessing the limits of determinism and precision in ultrafast laser ablation. Phys. Lett. 2020, 117, 171604.(France, Spain)
  7. Bliedtner, J.; Schindler, C.; Seiler, M.; Wachter, S.; Friedrich, M.; Giesecke, J. Ultrashort pulse laser material processing. Laser Technik Journal 2016, 5, 46-50.(Germany)
  8. Gedvilas, M.; Raciukaitis, G. Investigation of UV Picosecond Laser Ablation of Polymers. Proceedings. SPIE 6157, Workshop on Laser Applications in Europe2005, 61570T-1-11. (Lithuania)
  9. Choi, J; Schwarz C. Advances in femtosecond laser processing of opticalmaterial for device applications. Int J Appl Glass Sci. (International Journal of Applied Glass Science) 2020,11, 480–490. (USA)
  10. Mirza, I.; Bulgakova, N. M.; Tomáštík, J.; Michálek, V.; Haderka, O.; Fekete, L.; Mocek, T. Ultrashort pulse laser ablation of dielectrics: Thresholds, mechanisms, role of breakdown. Scientific Reports2016, 6, 39133. (Czech Republic; Russia)
  11. Huang, H.; Yang, L. M.; Liu, J. Femtosecond fiber laser based micro- and nano-processing.SPIE Optics and Photonics SPIE Paper Number: 8519-21 SPIE optics and Photonics August 2012, 12-16.(USA)
  12. Nolte, S; Momma C.; Jacobs, H; Tu¨nnermann, A.; Chichkov, B. N.; Wellegehausen, B.; Welling, H. Ablation of metals by ultrashort laser pulses. Opt. Soc. Am.B 1997, 14, 2716-2722. (Germany)
  13. Leitz, K. H.; Redlingshöfera, B.; Regc, Y.; Ottoa, A.; Schmidt, M. Metal ablation with short and ultrashort laser pulses. Physics Procedia2011, 12, 230–238. (Germany; Russia)
  14. Perry, M. D. B.; Stuart, C.; Banks, P. S.; Feit, M. D.; Yanovsky, V.; Rubenchik, A. M. Ultrashort-pulse laser machining of dielectric materials. Journal of Applied Physics1999, 85, 6803-6810. (USA)

Comment #5 of reviewer #3: Conclusions: could you give some of them as bullets, one per each highlight?

Response to comment #5 of reviewer #3: Thank reviewer for kind suggestions. The authors give some of conclusions as bullets, one per each highlight. Please refer to the following conclusions and  conclusions the revised manuscript. 

  1. Conclusions
  • The model proposed by this work can, respectively, divide the analysis of an ultrashort pulsed laser processing of materials into the optical and thermal effects using the Laplace transform 
  • The ablated depth per pulse versus laser fluence predicted by this study agreeswith the data measured by the published paper. Both optical and thermal ablations follow Beer’s exponential law based on optical absorption length and thermal diffusion length, respectively 
  • It has also been shown that optical and thermal ablation, respectively occur in low and high laser fluences because the optical ablation depends on the only main source of the direct incident ultrashort pulsed laser. However, the thermal ablation is governed by the residual heat of the directly incident ultrashort pulsed laser after the optical ablation. Therefore, the thermal ablation is more obvious for high laser fluences. If the laser fluence is not high enough, the thermal ablation is insignificant  
  • This paper analytically indicates that the depth per pulse of optical ablation approximates the product of optical absorption length and the logarithm of the ratio of laser fluence to the laser fluence threshold. The variation of ablation rate with laser fluence for data calculated by this work and measured by the published paper is shown in the coordinates with the horizontal axis of the logarithm scale and linear scale. In the coordinates with the horizontal axis of the logarithm scale, the rate of increase of the ablated depth per pulse with laser fluence is faster at higher laser fluences than that at low laser fluences. However, the rate of increase of the ablated depth per pulse with laser fluence is actually lower at high laser fluences than that at low laser fluences in the coordinates with the horizontal axis of linear scale because the thermal ablation using the residual heat of the directly incident ultrashort pulsed laser governs at high laser fluences

Comment #6 of reviewer #3: do you have some image of treated surfaces by microscopy?

Response to comment #6 of reviewer #3: Thank reviewer for kind suggestions. The authors have no images of treated surfaces by microscopy. In future studies, the authors will add some images of treated surfaces by microscopy into their manuscript.

Comment #7 of reviewer #3: Please tak your time to improve the paper. Last recommendation were accomplihed too ligthly.

Response to comment #7 of reviewer #3: Thank reviewers for valuable comments. The authors did their best to revise this manuscript according to the reviewers’ comments.
